# Doping-control of excitons and magnetism in few-layer CrSBr

Farsane Tabataba-Vakili [1,2] ✉, Huy P. G. Nguyen[1], Anna Rupp[1], Kseniia Mosina [3], Anastasios Papavasileiou[3], Kenji Watanabe [4], Takashi Taniguchi [5], Patrick Maletinsky [6], Mikhail M. Glazov [7], Zdenek Sofer [3], Anvar S. Baimuratov [1] ✉ & Alexander Högele [1,2] ✉

Magnetism in two-dimensional materials reveals phenomena distinct from bulk magnetic crystals, with sensitivity to charge doping and electric fields in monolayer and bilayer van der Waals magnet $CrI_3$. Within the class of layered magnets, semiconducting CrSBr stands out by featuring stability under ambient conditions, correlating excitons with magnetic order and thus providing strong magnon-exciton coupling, and exhibiting peculiar magneto-optics of exciton-polaritons. Here, we demonstrate that both exciton and magnetic transitions in bilayer and trilayer CrSBr are sensitive to voltage-controlled field-effect charging, exhibiting bound exciton-charge complexes and doping-induced metamagnetic transitions. Moreover, we demonstrate how these unique properties enable optical probes of local magnetic order, visualizing magnetic domains of competing phases across metamagnetic transitions induced by magnetic field or electrostatic doping. Our work identifies few-layer CrSBr as a rich platform for exploring collaborative effects of charge, optical excitations, and magnetism.

Recent experimental realization of two-dimensional (2D) magnets with ferromagnetic (FM) order down to the monolayer limit[1,2] has initiated extensive research on van der Waals magnets, with observation of magnons[3,4], magnetic proximity coupling[5,6], and giant magnetoresistance[7,8]. With additional electrostatic control of magnetism as in $CrI_3$[9–11], 2D magnets promise novel applications in spintronics or magnetic memories, including high-speed magnetic switching. More recently, antiferromagnetic (AF) semiconductor CrSBr with a bandgap of 1.5 eV[12,13], intralayer FM order and AF interlayer coupling[14], and high Néel temperatures of 132 and 140 K for bulk[15] and bilayer[16] crystals has received particular attention due to its intriguing magnetic[14,17] and magneto-optical properties[13,18], with demonstrations of strongly linearly polarized excitons sensitive to the magnetic order[18], magnon-exciton coupling[19,20], exciton-polaritons[21–23], and large negative magnetoresistance[12,24,25]. To date, however, electric control of magneto-optical phenomena in CrSBr has remained elusive.

In this work, we present an elaborate study of electrostatic control of the coupled excitonic and magnetic properties of few-layer CrSBr. To this end, we embed monocrystalline few-layer CrSBr in a field-effect device and perform cryogenic magneto-optical studies as a function of voltage-induced doping and in the presence of magnetic fields along the magnetic hard (crystallographic c), easy (b), and intermediate (a) axes. Upon electron doping, we observe the emergence of charged exciton complexes in bi- and trilayer crystals, and investigate their origins both experimentally and theoretically. The parabolic

[1]Fakultät für Physik, Munich Quantum Center, and Center for NanoScience (CeNS), Ludwig-Maximilians-Universität München, Geschwister-Scholl-Platz 1, 80539 München, Germany. [2]Munich Center for Quantum Science and Technology (MCQST), 80799 München, Germany. [3]Department of Inorganic Chemistry, University of Chemistry and Technology Prague, Technická 5, 166 28 Prague 6, Czech Republic. [4]Research Center for Functional Materials, National Institute for Materials Science, 1-1 Namiki, Tsukuba 305-0044, Japan. [5]International Center for Materials Nanoarchitectonics, National Institute for Materials Science, 1-1 Namiki, Tsukuba 305-0044, Japan. [6]Department of Physics, University of Basel Basel, Switzerland. [7]Ioffe Institute, 194021 Saint Petersburg, Russian Federation. ✉e-mail: f.tabataba@lmu.de; anvar.baimuratov@lmu.de; alexander.hoegele@lmu.de

dispersions in magnetic fields along the $c$ and $a$ axes[18] allow us to establish a self-consistent description of the neutral and charged exciton complexes in the presence of coupling between intralayer and interlayer excitons mediated by hole interlayer tunneling. We utilize this understanding to demonstrate electric control of the metamagnetic transitions induced by magnetic field along the $b$ axis, with pronounced exciton energy jumps correlated with the magnetic order in bi- and trilayer[18]. Finally, we demonstrate how the coupling between excitons and magnetism can be utilized for local sensing of magnetic phases, which depend on both magnetic field and charge doping, and extend the technique to optical raster-imaging of magnetic domains.

## Results and discussion

Our field-effect device incorporates a CrSBr flake with mono-, bi-, and trilayer regions, exfoliated from a bulk crystal grown by chemical vapor transport (details in Methods). The CrSBr flake shown in the optical micrograph of Fig. 1a has a characteristic shape, extended along the crystallographic $a$ axis[18,26,27]. Using the conventional dry-transfer method[28], we fabricated a single-gated device with hBN as dielectric and few-layer graphene as top gate and contact layer (see schematic in Fig. 1b, and Methods for fabrication details). To study the

sample by low-temperature differential reflectance (DR) and photoluminescence (PL) spectroscopy at 3.2 K as a function of electrostatic doping, we identify mono-, bi-, and trilayer regions with strong exciton resonances, marked by diamonds in the DR map of Fig. 1c.

The spectrum of the neutral monolayer at 0 V in the top panel of Fig. 1d shows a feature in DR at 1.342 eV, which corresponds to a broad absorption peak (see Supplementary Fig. 1a for absorption spectra determined by Kramers-Kronig relation) and low-intensity PL (top panel of Fig. 1e). The negligible energy shift between DR, absorption and PL indicates a momentum-direct exciton transition[18] labeled as $X_A$. Our theoretical analysis assigns the $X_A$ exciton to the transition between the topmost valence band ($v$) and the lowest conduction band ($c_1$) at the Γ point of the first Brillouin zone, without clear consensus on oscillator strength[13,14,17,18,29,30]. We adopt the notion of a nominally dipole-forbidden transition $v \leftrightarrow c_1$[13] (see Methods for details), brightened by the asymmetry of our structure and the high surface-to-volume ratio of the monolayer and persisting in the DR spectra of the neutral bi- and trilayer in the top panels of Fig. 1f, h, respectively (see Supplementary Fig. 2 for exciton layer assignment).

The assignment of the transition to the top valence and bottom conduction band is substantiated by the absence of striking signatures

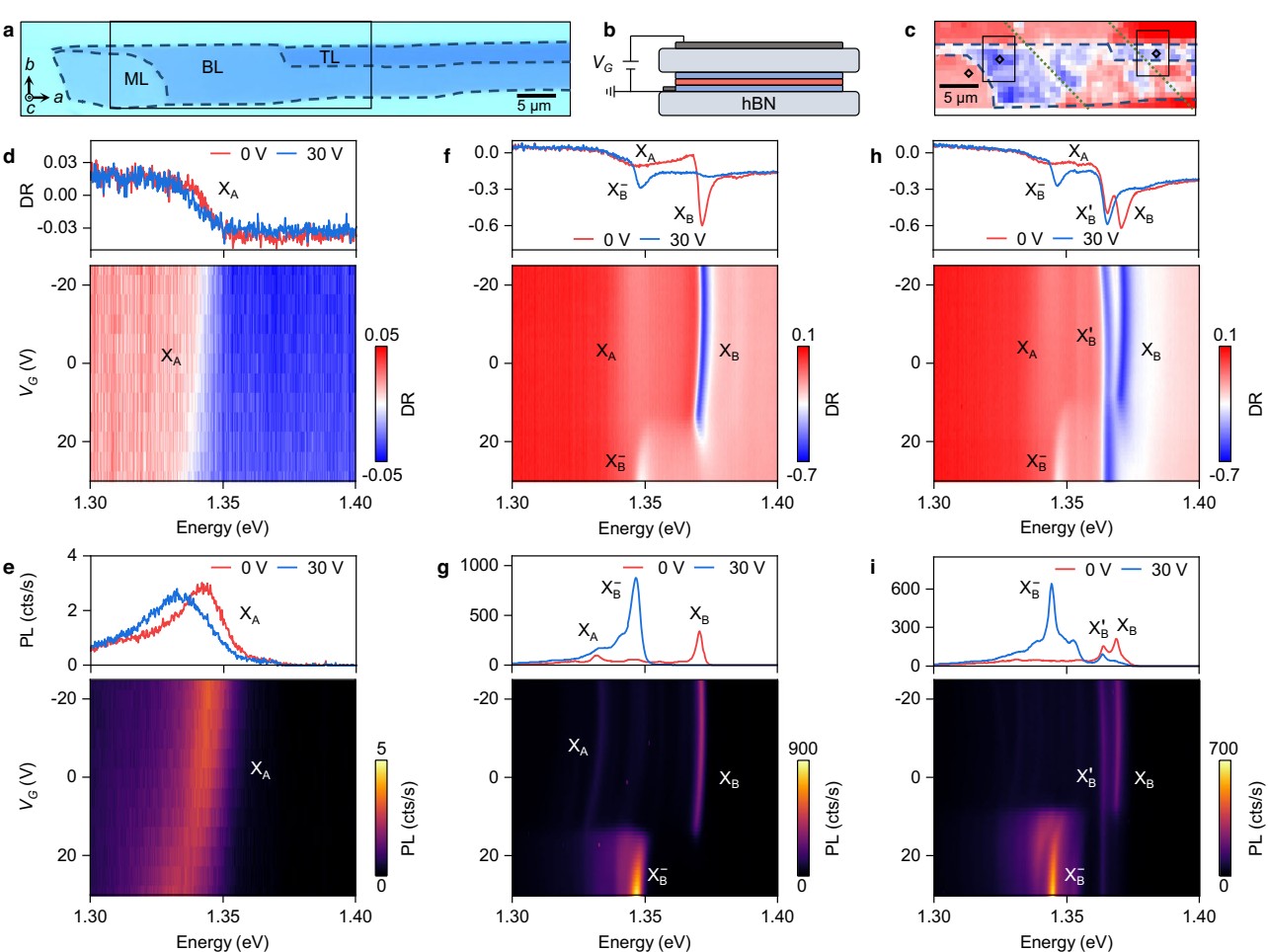

**Fig. 1 | CrSBr field-effect device and spectral signatures of doping. a** Optical micrograph of the few-layer CrSBr crystal, with crystal axes and layer numbers indicated (ML: monolayer, BL: bilayer, and TL: trilayer; dashed lines mark the crystal boundaries, the rectangle corresponds to the region in **c**). **b** Field-effect device layout with few-layer graphene top gate and contact (dark grey). The CrSBr layers (blue and red indicating spin polarization in antiferromagnetic order) are encapsulated by two hBN flakes (light grey). **c** Differential reflectance (DR) map of the region marked in **a** in the energy range of 1.3–1.4 eV (same color bar as **f, h**), with

blue areas corresponding to strong exciton resonances (dotted lines mark the few-layer graphene contact, and black diamonds indicate the positions on mono-, bi- and trilayer where all data were acquired; the two rectangles indicate the regions studied in Fig. 4). **d, e** Monolayer DR and photoluminescence (PL) spectra at 0 and 30 V (top panels) and the corresponding sweeps of the gate voltage $V_G$ (bottom panels), respectively. **f–i** Same as **d, e**, but for bilayer (**f, g**) and trilayer (**h, i**). $X_A$, $X_B$, and $X_B'$ label neutral and $X_B^-$ charged exciton transitions.

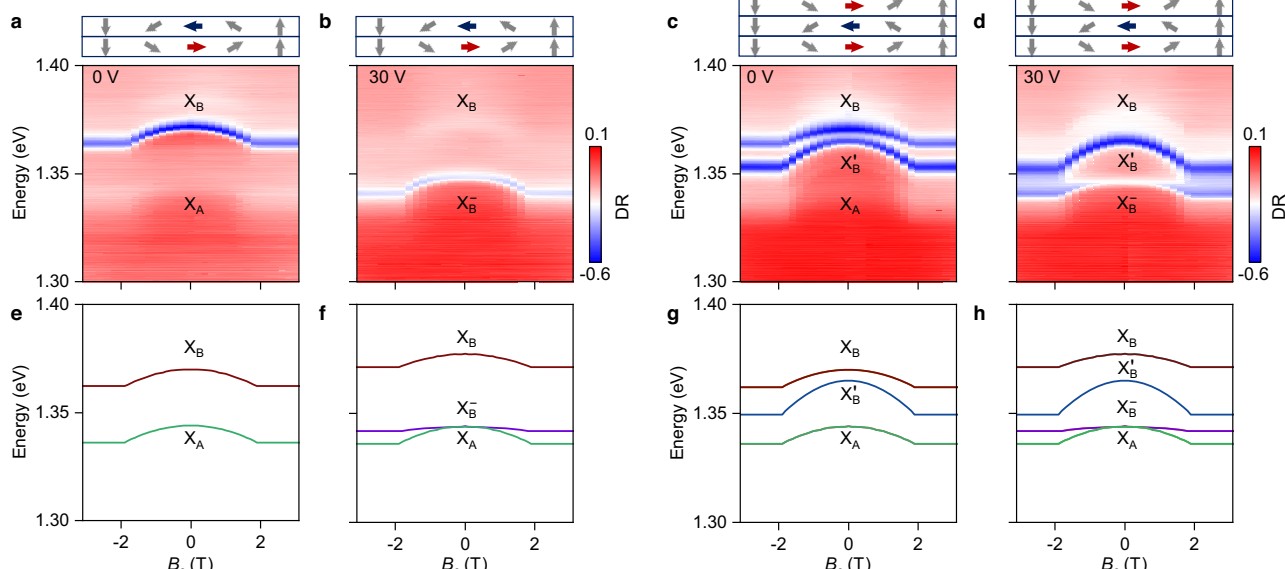

**Fig. 2 | Magneto-optical spectroscopy of neutral and charged bi- and trilayer in magnetic field along the $c$ axis. a, b** Evolution of bilayer differential reflectance (DR) with magnetic field along the $c$ axis ($B_c$) in the neutral and negatively charged regime at 0 and 30 V, respectively. The top panels illustrate the spin orientation, where blue and red colors indicate antiferromagnetic order with opposite spins, while all other spin orientations are grey. **c, d** Same as **a, b**, but for the trilayer. **e–h** Calculated dispersions of the corresponding neutral and charged excitons in magnetic field along the $c$ axis.

upon electrostatic doping in the FM monolayer, with data in Fig. 1d, e. Due to complete spin polarization of the bands, a bound exciton-charge state can not form with spin-aligned electrons, and we only observe a gradual red-shift of the $X_A$ resonance with the gate voltage in Fig. 1d, e, with a maximum shift of 7 meV at 30 V. Even though coupling of electrostatic field and doping effects can not be ruled out in our single-gated device and could account in part for the shift via the Stark effect, we observe clear signatures of doping in bi- and trilayer for the same voltage range both in DR (Fig. 1f, h) and PL (Fig. 1g, i).

The most striking spectral features of layer charging in Fig. 1f–i are the pronounced energy shifts at elevated positive gate voltages around 15 V, converting the spectrally narrow peaks $X_B$ with energy near 1.370 eV into $X_B^-$ peaks just below 1.350 eV. The energy difference of 19 meV, as well as the blue-shift of the $X_B$ resonance upon doping, are reminiscent of negatively charged trions[31,32] or Fermi-polarons[33–37] in monolayer transition-metal dichalcogenides, indicating enhanced Coulomb interactions due to reduced dielectric screening in the 2D semiconducting magnet. We assign the neutral exciton $X_B$ in Fig. 1f, h, with narrow absorption of ~3 meV full-width at half-maximum linewidth and negligible Stokes shift to the transition between the top valence band v and the higher-energy conduction subband $c_2$, which is dipole-allowed for linear polarization along the $b$ axis[13] (Supplementary Fig. 3 shows linearly polarized out-of-plane emission). The corresponding PL spectra in Fig. 1g, i stem from hot excitons, with incomplete relaxation of electron constituents from subband $c_2$ to $c_1$.

In the trilayer, the pronounced spectral feature just below $X_B$, labelled as $X'_B$ in Fig. 1h, i, deserves a separate discussion. The transition, red-shifted by 5.4 meV from $X_B$, is strong in DR and clearly present in PL, yet without signatures of doping. Our theoretical analysis (see Methods) identifies the corresponding exciton as being localized exclusively in the middle layer, where enhanced screening changes the energy of $X'_B$ excitons. Remarkably, the exciton shows no signature of charge-bound states (note the prevalence of its spectral features at voltages above the $X_B$ to $X_B^-$ cross-over in Fig. 1h, i). This observation indicates that excitons in the middle layer are not subjected to intra-layer charge. The top and bottom layers of the trilayer host field-induced electrons in their lowest-energy $c_1$ states, downshifted by the magnetic layer interactions with respect to the $c_1$ band of the middle

layer. We note that a second sample reproduces all discussed neutral and charged exciton transitions in mono-, bi-, and trilayers (see Supplementary Fig. 4).

To elucidate the rich exciton multiplicity in bi- and trilayer crystals, we performed magneto-optical studies in magnetic fields along the $c$ axis (perpendicular to the layers), and analyzed the respective exciton dispersions in the framework of the states introduced above. The data in Fig. 2a, b show the evolution of DR with magnetic field for the neutral and charged bilayer at 0 and 30 V, respectively, with the corresponding data for the trilayer shown in Fig. 2c, d. We first note that all states involved exhibit symmetric energy red-shifts from 0 T towards increasing absolute values of the magnetic field, before they level off beyond a saturation field of ~2 T[13,18,38]. This behavior can be understood by considering spin canting from the AF state at 0 T to the FM state at sufficiently high magnetic fields[18,38], as indicated by the arrows in the top panels of Fig. 2a–d.

In the neutral bilayer with data in Fig. 2a, the energies of $X_A$ and $X_B$ reduce parabolically with the same dispersion to settle in the saturated FM regime with a broader linewidth and nearly preserved contrast in DR. The same magneto-dispersion of both excitons is consistent with different conduction bands and a shared valence band. In the electron-doped limit at 30 V of Fig. 2b, $X_B^-$ exhibits a parabolic energy red-shift similar to $X_B$ with the same saturation field, but the dispersion is flatter in the vicinity of 0 T. The magneto-dispersions of $X_A$, $X_B$, and $X_B^-$ in the neutral and charged trilayer (with data in Fig. 2c, d) are similar to the bilayer, whereas the curvature of the dispersion of $X'_B$ is twice as large and unaffected by doping. The magneto-dispersions along the $a$ axis reproduce these observations with a lower saturation field of 1 T (see Supplementary Figs. 5 and 6).

Such negative diamagnetic shifts of neutral and charged exitons are highly unusual in conventional semiconductors[39]. They can be understood in the framework of our analysis, invoking interlayer excitons with smaller binding energy and, consequently, higher absolute energy[40], in addition to intralayer excitons and their charged counterparts introduced above. With magnetic field along the $c$ axis causing gradual spin canting from $b$ axis AF to $c$ axis FM order, spin-conserving hole tunneling sets in to mix intra- and interlayer exciton states. Our analysis (see Methods) shows in Fig. 2e that the magneto-

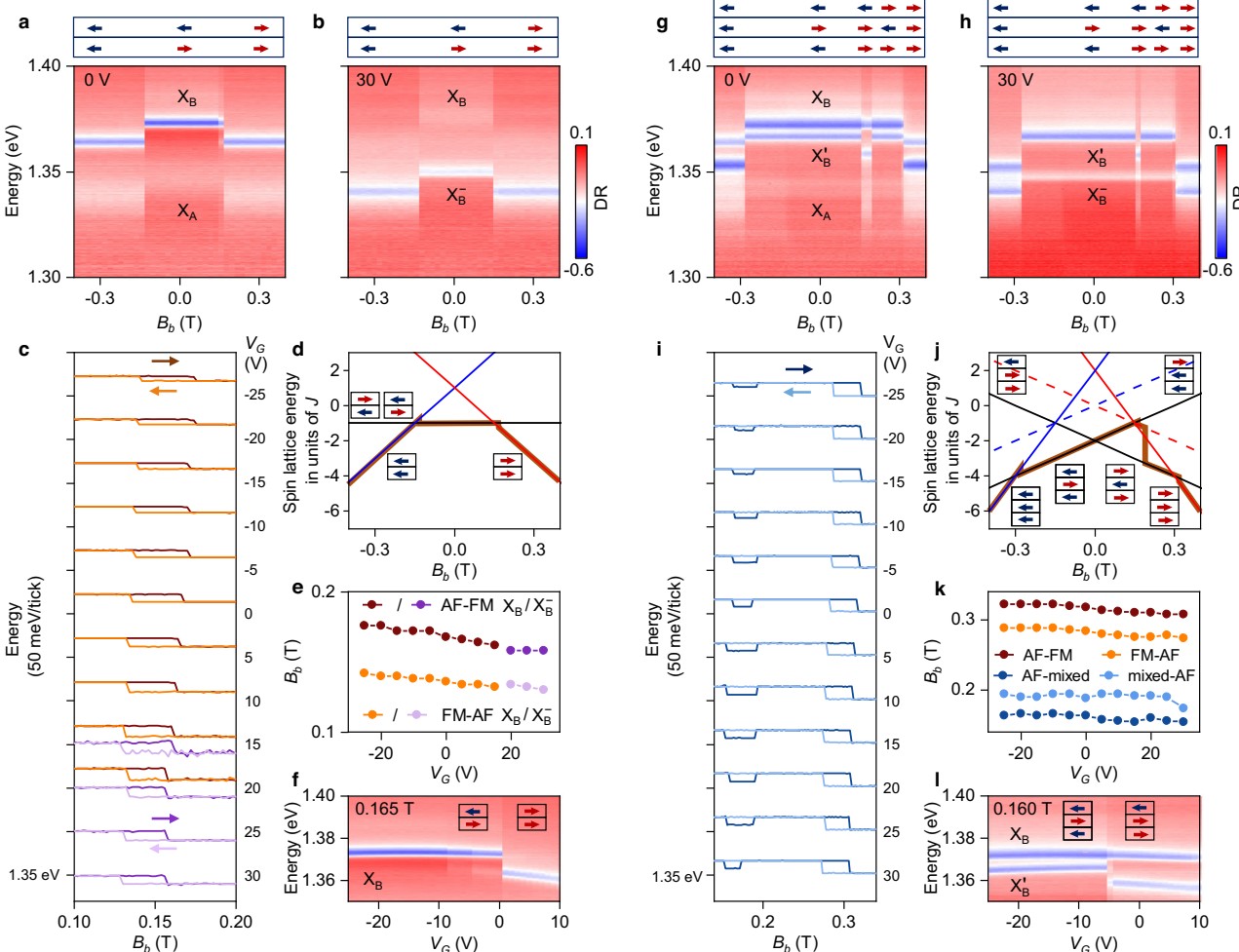

**Fig. 3 | Effect of doping on excitons and metamagnetic transitions in magnetic field along the *b* axis. a, b** Differential reflectance (DR) as a function of increasing magnetic field along the *b* axis ($B_b$) at 0 and 30 V, respectively. The top panels indicate the bilayer spin orientation. **c** Energy of $X_B$ (brown/orange) and $X_B^-$ (dark/light purple) at different gate voltages for upward and downward magnetic field sweep directions indicated by right and left arrows in the corresponding colors, respectively. The hysteresis loops at each gate voltage $V_G$ are offset by 50 meV for clarity; note that both $X_B$ and $X_B^-$ coexist at 15 and 20 V. **d** Calculated bilayer spin lattice energy as a function of $B_b$ for the antiferromagnetic (AF, black line) and ferromagnetic (FM, blue and red lines) phases indicated by the arrows. The brown

line shows the upward sweep with jumps corresponding to the experimental data in **a**. **e** Evolution of the critical magnetic field of the AF-FM metamagnetic transition with gate voltage. **f** Doping-induced metamagnetic switching from AF to FM at a magnetic field of 0.165 T, with arrows indicating spin orientations. **g–l** Same as **a–f**, but for the trilayer. **i** Hysteresis loops of $X_B'$. **j** Spin lattice energy for trilayer AF (solid black), FM (solid blue/red), and mixed (dashed blue/red) phases. **k** Critical magnetic fields of the metamagnetic transitions observed in the trilayer. **l** Doping-induced switching in the trilayer from AF to mixed state at a magnetic field of 0.160 T. Blue and red colored arrows in **a**, **b**, **d**, **g**, **h**, **j**, **l** illustrate layers with left and right spin orientation, respectively.

dispersions of $X_{A,B}$ excitons can be represented for magnetic fields $B < B_{sat}$ (with the saturation field $B_{sat}$ corresponding to the FM order) as $\mathcal{E}_{A,B}(B) = \mathcal{E}_{A,B}(0) - tB^2$, with the fitting parameter $t \approx 2$ meV/T² related to the interlayer hole tunneling and the energy between the intralayer and interlayer excitons.

Within the minimal model of Fermi-polarons (see Methods for details), where we account for the Fermi-sea mediated coupling of intralayer excitons and trions[36,37], the magneto-dispersion of $X_B^-$ in Fig. 2f is weaker: it contains an additional factor of $E_F/E_{tr}$, given by the ratio of the Fermi energy $E_F$ to the trion binding energy $E_{tr}$. With $E_F = 10$ meV and $E_{tr} = 19$ meV extracted from the data in Fig. 1, the model predicts a slightly flatter trion dispersion than observed in the experiment of Fig. 2b, possibly because of disregarded intra-interlayer trion coupling. The dispersions in the neutral and charged trilayer can be understood along the same lines, as shown in Fig. 2g, h for the same set of fit parameters, by taking into account that $X_B$ stems from the outer layers and $X_B'$ exclusively resides in the middle layer. The main difference is that hole tunneling to both the top and bottom layers is not suppressed, which results in $\mathcal{E}_B'(B) \approx \mathcal{E}_B'(0) - 2tB^2$ with twice as

large curvature in the dispersion of $X_B'$ as compared to $X_{A,B}$, in full agreement with experimental data.

The gradual spin canting transition along the *c* axis is contrasted by an abrupt metamagnetic spin-flip transition[18,41] along the *b* axis parallel to the magnetic moment. At a critical magnetic field near 0.17 T, the spins flip from AF to FM order in both the neutral and charged bilayer, when swept upward from negative to positive values as in Fig. 3a, b. Unlike in magneto-dispersions along the hard *c* axis, the critical field along the easy axis depends hysteretically on the sweep direction (Fig. 3c), which is indicative of a first-order magnetic transition well known from optical spectroscopy[18] and magneto-transport studies[12,24,25].

The critical field is determined by the spin-lattice energy, which in turn is governed by the interplay between the interlayer exchange energy *J* and magnetic field interaction[24]. From the data, we estimate the spin-lattice energy for AF and FM states as a function of the magnetic field, as shown in Fig. 3d, assuming the crossing points between FM and AF states at the center of the hysteresis (see Methods for details). In the upward direction and with initialization in the FM state

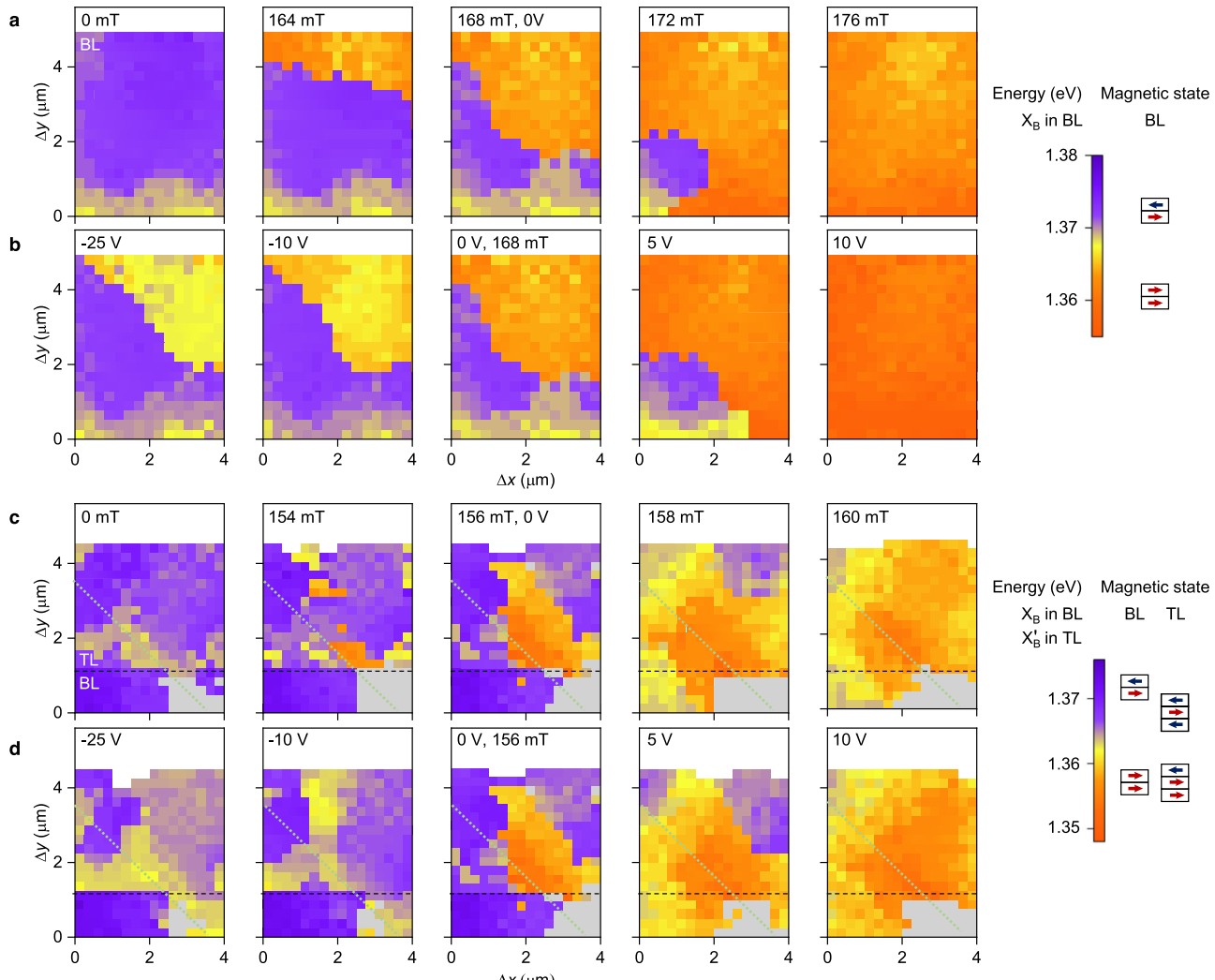

**Fig. 4 | Optical imaging of magnetism in bi- and trilayer CrSBr. a** Raster-scan maps of a bilayer (BL) region (left rectangle in Fig. 1c; the white areas delimit the crystal top edge) with bias at 0 V for five consecutive magnetic fields along the *b* axis, with false-color contrast given by the energy of the $X_B$ transition in differential reflectance (DR) with corresponding antiferromagnetic (AF, purple) and ferromagnetic (FM, orange) phases indicated on the right of the color bar. **b** Same as **a**, but for a constant field of 168 mT and five different gate voltages. **c, d** Equivalent maps of the trilayer (TL) region (right rectangle in Fig. 1c, with the dashed and dotted lines indicating the bi- to trilayer boundary and the edge of the graphene contact, respectively; grey pixels correspond to vanishing exciton DR due to local disorder) for magnetic fields across the metamagnetic one-flip transition (**c**) and voltage-controlled magnetism at a constant field of 156 mT (**d**). In all panels, the pixel color derives from the energies of $X_B$ in the bilayer and $X'_B$ in the trilayer, respectively, and the corresponding magnetic states are indicated on the right of the color bar, with blue and red arrows representing layers with left and right spin orientation, respectively.

at a negative magnetic field, the spins flip from FM to AF order and back to FM, as illustrated by the brown trace in Fig. 3d extracted from the data in Fig. 3a. The hysteretic behavior near AF-FM degeneracy is responsible for abrupt jumps in the brown trace of Fig. 3d. Crucially, our device allows to tune the critical field of the spin-flip transition with gate voltage, as evident from the data in Fig. 3c, e. This effect, also observed in bilayer CrI₃[9,10], is consistent with an inverse scaling of the interlayer exchange energy with doping, which reduces the critical field upon negative doping. According to the Kugel-Khomskii model, the interlayer exchange energy is inversely proportional to the on-site Coulomb interaction[42], which in turn is proportional to the electron density. This implies that the metamagnetic AF-FM transition near the critical field can be equivalently induced by doping, as demonstrated in Fig. 3f.

The neutral and charged trilayer exhibit a similarly abrupt AF-FM transition as the bilayer, however, at larger critical fields near 0.32 T (Fig. 3g, h). Surprisingly, we also observe an additional transition into an intermediate or mixed state near 0.16 T, which is stable for 30 mT

before switching back to AF order. In this regime, the $X'_B$ resonance first jumps to lower energy (still above its energy in the FM state) and then back, while the energy of $X_B$ remains unaffected. This intricate signature corresponds to mixed order in the trilayer, resulting in two adjacent layers with FM and AF order illustrated in the top panel of Fig. 3g.

The corresponding spin-lattice energies for AF, FM, and mixed states shown in Fig. 3j provide some intuition for the spin-flip transitions as a function of the magnetic field. In the upward direction, with initialization in the FM state at a negative magnetic field, the trilayer is in an energetically favorable FM state (solid blue line). With increasing magnetic field, the spins of the middle layer flip to AF order, which is preserved beyond 0 T as a metastable state, where its time-reversal counterpart would be energetically more favorable. An energy-reducing transition, however, would require the simultaneous spin-flip in all three layers. With increasing field, the spin-lattice energy of the AF state crosses the FM and the mixed states (solid and dashed red line, respectively), where the trilayer transitions into a metastable

mixed state. Eventually, as this state becomes less favorable with increasing magnetic field, a double spin-flip occurs into the energetically favorable AF order (solid black line) and finally into the FM state (solid red line). Overall, this intricate sequence of spin-flips in the trilayer with magnetic field is driven by the minimization of the energy costs for spin-flips in magnetically ordered layers. As in the bilayer, the critical fields of the respective transitions in the trilayer depend on the charge doping, as evident from Fig. 3i, k, l for the AF-FM, FM-AF, and AF-mixed transitions, shifting towards lower values with increasing doping. Analogously to the bilayer, the metamagnetic transitions can also be induced by doping, as shown explicitly for the AF-mixed state transition in Fig. 3l.

With the observation that the exciton energy is correlated with the underlying magnetic order as in Figs. 2 and 3, the exciton transitions can be employed for all-optical detection of local magnetic order. In the following, we demonstrate this feature by mapping out domains of magnetic phases and their sensitivity to doping near metamagnetic transitions. We start out with the bilayer, initialized in its AF ground state at zero magnetic field and zero doping (leftmost panel of Fig. 4a), imaged by hyperspectral raster-scan DR mapping with purple and orange colors representing AF and FM domains, respectively. With increasing field, the AF domain shrinks, and the FM domain emerges near the critical field at the top edge of the crystal (maps of Fig. 4a at consecutive fields of 164, 168, and 172 mT), until full coverage is reached at 176 mT (rightmost panel of Fig. 4a), with the bottom left corner next to the monolayer flipping last. Remarkably, the characteristic local nucleation and propagation of FM domains can be induced by doping, as highlighted for five increasing gate voltages in Fig. 4b at a constant field of 168 mT.

We use the technique to visualize the magnetic transition from the AF to the mixed state in Fig. 4c (see also Supplementary Fig. 7 for large-area mapping of all trilayer transitions). With increasing magnetic field, the mixed state nucleates at the edge of the few-layer graphene contact and spreads along the $a$ axis ($x$ direction). We note that the AF-mixed transition in the trilayer and the AF-FM transition in the adjacent bilayer happen at the same field within the resolution limit, which indicates that the flip in the trilayer is induced by intralayer exchange coupling to the proximal bilayer. Remarkably, the trilayer mixed state is clearly discerned from the AF state despite nominally identical magnetization, which renders their discrimination difficult with conventional methods[12,16,18]. As in the bilayer, related phenomena of phase transitions can be induced by doping and detected all-optically, as demonstrated in Fig. 4d.

Our work identifies few-layer CrSBr as an intriguing platform to study and control excitons and magnetism with electrostatic doping. The rich phenomena observed are intertwined, providing a route to novel devices involving not only coupled electric and magnetic phenomena, but also adding optical means to control and exploit magnetism via neutral and charged excitons. The aspects highlighted in our work indicate the existence of rich magnetic phases in trilayer crystals beyond just AF or FM order, which can potentially be utilized in novel devices for spintronics and magnetic memories featuring layer-selective initialization, control, and read-out by combined electrostatic and optical means. Finally, the features cooperatively manifest optical imaging of magnetic order in CrSBr as an efficient and sensitive local probe of magnetic domains, providing insight complementary to other techniques[43-45].

## Methods

### CrSBr synthesis
CrSBr bulk single crystals were synthesized by chemical vapor transport method using chromium (99.99%, −60 mesh, Chemsavers), sulfur (99.9999%, 1–6 mm Wuhan Xinrong New Material Co. Ltd), and bromine (99.9999%, Sigma Aldrich), combined with a stoichiometry of 1:1:1, sealed in a quartz tube under high vacuum, and then placed into a two-zone tube furnace. The material was pre-reacted in an ampoule at 700 °C for 25 h until most of the bromine was reacted. During this procedure one part of the ampoule was kept under 200 °C to avoid pressure disruption of the ampoule. The crystal growth by the vapor transport method was performed in a two-zone horizontal furnace. First, the source and growth ends were kept at 800 and 900 °C, respectively. After 25 h, the temperature gradient was reversed, and the temperature at the hot end was gradually increased from 880 to 930 °C for an 8-day period while the growth zone was kept at 800 °C. The high-quality CrSBr single crystals were removed from the ampoule in an Ar glovebox.

### Field-effect devices
CrSBr, few-layer graphene, and hBN (NIMS) flakes were exfoliated from bulk single crystals at ambient conditions onto SiO₂/Si substrates. Suitable CrSBr flakes were identified by optical contrast and atomic force microscopy. A PDMS/PC stamp was used to sequentially pick up the hBN, few-layer graphene, and CrSBr layers employing the dry-transfer method[28]. Poly-(Bisphenol A-carbonate) pellets (Sigma Aldrich) were dissolved in chloroform with a weight ratio of 6. The mixture was stirred overnight at room temperature at 500 rpm using a magneton bar. The well-dissolved PC film was mounted on a PDMS dome on a glass slide. First, the top hBN layer with a thickness of 64 nm was picked up with the stamp, followed by the CrSBr flake, a few-layer graphene contact layer, and 84 nm bottom hBN. The stack was released at a temperature of 190 °C onto a pre-patterned SiO₂/Si target substrate with Ti/Au metal pads, then soaked in chloroform solution for 2 min to remove PC residues and cleaned by acetone and iso-propanol. Next, the top gate few-layer graphene flake was picked up and placed on top of the heterostack, followed by another cleaning step. The pick-up temperature for CrSBr was around 110 °C, for the other flakes -100 °C. The sample was annealed at 200 °C under ultra-high vacuum for 15 h. Then, Ti/Au contact stripes were fabricated to connect the few-layer graphene gates to the contact pads using laser lithography and electron-beam evaporation. The second sample was fabricated in the same way but using hBN flakes with 28.5 and 26 nm for the top and bottom encapsulating layers, respectively. Electrostatic doping was controlled by applying a gate voltage with a programmable DC-source (Yokogawa7651) between the gate and the grounded contact layer.

### Magneto-optical spectroscopy
Cryogenic PL and DR spectroscopy were performed in back-scattering geometry with a lab-built confocal microscope in a close-cycle cryostat (attocube systems, attoDRY1000) with a base temperature of 3.2 K and a solenoid with magnetic fields of up to ±9 T. Magnetic field sweeps along the $b$ axis in upward (downward) direction were performed by initializing the magnet at −500 mT (500 mT) and then ramping to the target field. We estimate the magnetic field inaccuracy to 2 mT for sweeps in steps of 2 mT, as deduced from repeated measurements under nominally identical conditions. Measurements with magnetic field along the $c$ axis were conducted by positioning the sample with respect to a low-temperature apochromatic objective (attocube systems, LT-APO/NIR/0.81) with piezo-units (attocube systems, ANPx101, ANPz101, and ANSxy100). For measurements along the $b$ and $a$ axes, a custom-built Voigt adapter was used, consisting of a mirror mounted at 45° and an aspheric lens (Geltech 350330) glued onto a Ti part. The sample holder was mounted on an L-shaped adapter with the crystallographic $b$ or $a$ axis of the sample aligned with the axis of the solenoid. The L-shaped adapter was mounted on piezo-units (attocube systems, ANPx101, ANPz101, and ANSxyz100) for nanopositioning and scanning. Momentum-space imaging in 4f and telescope configuration employed four achromatic doublet lenses (Edmund Optics, VIS-NIR) and was performed in an attoDRY800 close-cycle cryostat with 4 K base temperature.

In experiments on sample 1, PL was excited at 870 nm and 100 µW with a Ti:sapphire laser (Coherent, Mira) in continuous-wave mode and Semrock tunable short-pass and long-pass filters were used (887 nm VersaChrome Edge) in excitation and detection. DR, defined as DR = $(R - R_0)/R_0$, where $R$ was the reflectance from the sample and $R_0$ was the reference reflectance on the nearby substrate with hBN, was recorded with a Tungsten-Halogen lamp (Thorlabs, SLS201L or Ocean Insight, HL-2000-HP). The signal was dispersed by a monochromator (Roper Scientific, Acton SpectraPro 300i or Acton SP250 or Teledyne Princeton Instruments, IsoPlane SCT320) with a 300 grooves/mm grating and detected by a Peltier-cooled (Andor, iDus 416 or Teledyne Princeton Instruments, PIXIS 1024) or liquid nitrogen-cooled (Spec-10:100BR) charge-coupled device. A set of linear polarizers (Thorlabs, LPVIS), half- or quarter-waveplates (B. Halle, 310–1100 nm achromatic) mounted on piezo-rotators (attocube systems, ANR240) were used to control the polarization in excitation and detection. For sample 2, excitation and detection were circularly polarized, and PL was excited at 800 nm and 100 µW with a Ti:sapphire laser (SolsTiS, M Squared) with 842 nm short-pass (Semrock 842/SP BrightLine HC Short-pass Filter) and 808 nm long-pass (Semrock LP Edge Basic Long-pass Filter) filters in excitation and detection.

## Fermi-polaron model

We develop the Fermi-polaron model for multilayer CrSBr based on the approach used for 2D semiconductors[35–37], taking into account the spin-polarized band structure of CrSBr. We recall that both monolayer and multilayer CrSBr are described by a centrosymmetric $D_{2h}$ point symmetry group. According to Ref.[13], in the set of axes with $z \| c$ (normal to the monolayer, magnetic hard axis), $y \| b$ (magnetic easy axis), and $x \| a$ (magnetic intermediate axis) the orbital Bloch function of the topmost valence band $v$ (we account for a single valence band $v$ owing to its significant separation from the lower-lying bands) in the monolayer is transformed according to the $B_{3g}$ (or $\Gamma_4^+$, i.e., as $yz$) irreducible representation, and for two nearest conduction bands $c_1$ and $c_2$ according to $A_g$ ($\Gamma_1^+$ as $x^2 + y^2 + z^2$) and $B_{1u}$ ($\Gamma_3^-$ as $z$), respectively. Intralayer FM spin-spin interactions result in the complete spin polarization of the Bloch states with the spins aligned along the $b$ ($y$) axis in the monolayer. In multilayers, spins are aligned antiferromagnetically along the positive and negative directions of the $b$ axis in neighboring layers. Optical transition $v \leftrightarrow c_1$ is forbidden in the dipole approximation and the transition $v \leftrightarrow c_2$ is allowed for light polarized along the easy axis $b \| y$. Note that the predicted order of $c_1$ and $c_2$ bands varies in the literature, cf. Refs. 13,18,46, due to the small $c_1$–$c_2$ energy splitting and, correspondingly, its strong dependence on the ab initio and model parameters.

The lowest-energy optical transition in the monolayer, labeled as $X_A$ is linearly polarized along the $b \| y$ axis, broad in absorption and faint in PL, without a sizable Stokes shift. The latter feature indicates a momentum-direct exciton transition, which, however, should be nominally forbidden by dipolar selection rules. We speculate that due to the asymmetry of the structure (inequivalence of $z \to -z$) enhanced by the large surface-to-volume ratio in the monolayer, the dipolar selection rules are compromised and the $v \leftrightarrow c_1$ transition becomes weakly allowed. $X_A$ also prevails in the spectra of the bi- and trilayer, where its features in absorption and PL are contrasted by the much more pronounced and spectrally narrow resonances of $X_B$, which we assign to the dipole-allowed $v \leftrightarrow c_2$ transition with linear polarization along the $b \| y$ axis.

With this understanding, we consider the manifold of relevant neutral and negatively charged exciton states as in the Supplementary Fig. 2. We assume that the difference of the binding energies of the intra- and interlayer excitons exceeds by far the $c_1$–$c_2$ conduction band splitting. We also neglect bound charge complexes of interlayer excitons (negative interlayer trions) due to much smaller binding energies[40]. Furthermore, we neglect interlayer electron and hole

tunneling at zero magnetic field where single-particle tunneling between the same bands is spin-forbidden, while at finite fields the tunneling $c_1 \leftrightarrow c_2$ is suppressed by the conduction band splitting. At finite magnetic field along the $c$ axis, the spins in the neighboring layers are canted, and hole tunneling between the layers becomes allowed[18].

In this framework, the bilayer system Hamiltonian breaks into two identical blocks describing electrons either in the top or in the bottom layer:

$$H_{\mathrm{BL}} = \begin{pmatrix} H & 0 \\ 0 & H \end{pmatrix}. \tag{1}$$

Each block accounts for both $vc_1$ intra- and interlayer (A) excitons and $vc_2$ intra- and interlayer (B) excitons, as well as their coupling with the corresponding Fermi-sea in the bottom conduction subband $c_1$:

$$H = \begin{pmatrix} E_{\mathrm{A}} & sB & 0 & 0 & 0 \\ sB & E_{\mathrm{IA}} & 0 & 0 & 0 \\ 0 & 0 & E_{\mathrm{B}} & sB & \sqrt{E_{\mathrm{tr}} E_{\mathrm{F}}} \\ 0 & 0 & sB & E_{\mathrm{IB}} & 0 \\ 0 & 0 & \sqrt{E_{\mathrm{tr}} E_{\mathrm{F}}} & 0 & E_{\mathrm{B}} - E_{\mathrm{tr}} \end{pmatrix}, \tag{2}$$

where $E_{\mathrm{A(B)}}$ are the bare energies of A(B)-excitons (in absence of tunneling and electron doping); $E_{\mathrm{IA(IB)}} = E_{\mathrm{A(B)}} + \Delta$ are the energies of the interlayer A(B)-excitons with the difference in the binding energies $\Delta$. The coefficient $s$ quantifies the magneto-induced interlayer hole tunneling; $E_{\mathrm{F}}$ is the Fermi level and $E_{\mathrm{tr}}$ is the trion binding energy. The Hamiltonian in Eq. (2) corresponds to the non-self-consistent approximation for the exciton-electron scattering matrix element, and we set the coupling parameter to be $\sqrt{E_{\mathrm{tr}} E_{\mathrm{F}}}$, omitting a numerical coefficient -1 as well as the Fermi-sea effect on the trion binding energy[47].

In the weak doping regime ($E_{\mathrm{F}} \ll E_{\mathrm{tr}}$) and moderate magnetic fields ($|sB| \ll \Delta$), we obtain the energies of the A-excitons (decoupled from the Fermi-sea) and Fermi-polarons formed by B-excitons in the form:

$$\mathcal{E}_{\mathrm{A}}(B) \approx E_{\mathrm{A}} - tB^2, \tag{3a}$$

$$\mathcal{E}_{\mathrm{B}}^{\mathrm{RP}}(B, E_{\mathrm{F}}) \approx E_{\mathrm{B}} - tB^2 + E_F, \tag{3b}$$

$$\mathcal{E}_{\mathrm{B}}^{\mathrm{AP}}(B, E_{\mathrm{F}}) \approx E_{\mathrm{B}} - E_{\mathrm{tr}} - E_{\mathrm{F}} - \frac{E_{\mathrm{F}}\Delta}{E_{\mathrm{tr}}(\Delta + E_{\mathrm{tr}})} tB^2, \tag{3c}$$

with $t = s^2/\Delta$. In this regime, the A-exciton and B-exciton (repulsive polaron, RP) have the same magneto-dispersion, while the attractive Fermi-polaron (AP) has a smaller dispersion in magnetic field due to the additional term $E_{\mathrm{F}}/E_{\mathrm{tr}} \ll 1$. In the main text we use the notation $\mathcal{E}_{\mathrm{B}}(B) = \mathcal{E}_{\mathrm{B}}^{\mathrm{RP}}(B, 0)$.

Due to a sizable spread in the calculated conduction band splitting $c_1$–$c_2$[13,18], we take $E_{\mathrm{A,B}}$ as two fitting parameters. By fitting the experimental data to the eigenvalues of the Hamiltonian in Eq. (2), we obtain $E_{\mathrm{A}} = 1344$ meV, $E_{\mathrm{B}} = 1370$ meV, $t = 2.2$ meV/T², $B_{\mathrm{sat}} = 1.9$ T, $E_{\mathrm{tr}} = 19$ meV, and $E_{\mathrm{F}} = 10$ meV. In Fig. 2e, f, we show the eigenenergies of $X_A$, $X_B$, and $X_B^-$ in the neutral and negatively charged regime, respectively.

In the trilayer, we also assume suppressed interlayer tunneling of electrons as well as the exciton as a whole quasiparticle. The relevant states are shown in Supplementary Fig. 2c, and the Hamiltonian is written as:

$$H_{\mathrm{TL}} = \begin{pmatrix} H & 0 & 0 \\ 0 & H' & 0 \\ 0 & 0 & H \end{pmatrix}, \tag{4}$$

where the blocks of the states related to the top pair of layers and the bottom pair of layers are identical to the bilayer Hamiltonian in Eq. (2), while the Hamiltonian $H'$ is different, describing electrons in the middle layer.

We assume that the states in the middle layer have a different energy due to a different dielectric environment as compared to the outer layers, and label such excitons with a prime (see Supplementary Fig. 2c). Based on experimental observations (Figs. 1h and 2c) that (i) $X'_B$ has no charge-bound state upon doping, and (ii) no clear signature of $X'_A$ is observed, we disregard these states in our model (Supplementary Fig. 2c). Consequently, the middle block $H'$ in the Hamiltonian of Eq. (4) reduces to:

$$H' = \begin{pmatrix} E'_B & sB & sB \\ sB & E'_{IB} & 0 \\ sB & 0 & E'_{IB} \end{pmatrix}, \tag{5}$$

where $E'_{IB} = E'_B + \Delta$ is the energy of the interlayer B-excitons in the middle layer (note the multiplicity of two for such states).

Thus, the spectrum of the trilayer is a superposition of the bilayer spectrum (Eqs. (3)) and the spectrum of the middle layer that for $|sB| \ll \Delta$ approximates to:

$$\mathcal{E}'_B(B) \approx E'_B - 2tB^2. \tag{6}$$

We note that the curvature of its dispersion in magnetic field is twice that of the bilayer excitons described by Eq. (3b), as the two interlayer excitons couple with the intralayer exciton. By extracting the splitting at 0 T (AF state) in the neutral regime (as in Figs. 1h and 2c) we determine $E'_B = 1365$ meV. The eigenenergies of $X_A$, $X_B$, $X'_B$, and $X^-_B$ in the neutral and negatively charged regimes are shown in Fig. 2g, h, respectively.

### Estimation of the spin lattice energy

To understand the spin-flip transitions in magnetic field along the $b$ axis as in Fig. 3a, g, we estimate the spin-lattice energy by the following Hamiltonians for bi- and trilayers[42]:

$$H^{(s)}_{BL} = Js_1s_2 + \mu B_b(s_1 + s_2), \tag{7}$$

$$H^{(s)}_{TL} = J(s_1s_2 + s_2s_3) + \mu B_b(s_1 + s_2 + s_3), \tag{8}$$

where $B_b$ is the magnetic field along the $b$ axis, $s_{1,2,3} = \pm 1$ are the spin orientations along the $b$ axis for the respective layers, $J$ is the positive interlayer exchange between neighboring layers, and $\mu$ is the magnetic dipole moment of one layer. We assume that each layer exhibits intralayer FM order and neglect the intralayer exchange energies in the Hamiltonians of Eq. (7) and (8). The calculations are shown in Fig. 3d, j for bi- and trilayer, respectively.

## Data availability

The data that support the findings of this study are available at https://doi.org/10.5282/ubm/data.450[48].

## Code availability

The codes that support the findings of this study are available from the corresponding authors upon request.

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

## Acknowledgements

We thank J. Förste for help with the Python control of the setup. This research was funded by the European Research Council (ERC) under grant agreement no. 772195 and the Deutsche Forschungsgemeinschaft (DFG, German Research Foundation) within Germany's Excellence Strategy under grant no. EXC-2111-390814868. F.T.-V. acknowledges funding from the Munich Center for Quantum Science and Technology (MCQST) and the European Union's Framework Programme for Research and Innovation Horizon Europe under the Marie Skłodowska-Curie Actions grant agreement no. 101058981. A.R. acknowledges funding by the Bavarian Hightech Agenda within the Munich Quantum Valley doctoral fellowship program. A.P. was supported by the Onassis Foundation Scholarship ID: F ZS 045-1/2022-2023 and Bodossaki Foundation Scholarship. K.W. and T.T. acknowledge support from JSPS KAKENHI (grant No. 19H05790, 20H00354 and 21H05233). P.M. acknowledges financial support from the ERC consolidator grant project QS2DM and SNF project no. 188521. Theoretical analysis by M.M.G. was supported by RSF project 23-12-00142. Z.S. was supported by the ERC-CZ program (project LL2101) from the Ministry of Education, Youth and Sports (MEYS). A.S.B. acknowledges funding by the European Union's Framework Programme for Research and Innovation Horizon 2020 under the Marie Skłodowska-Curie grant agreement no. 754388 (LMUResearchFellows) and from LMUexcellent, funded by the Federal Ministry of Education and Research (BMBF) and the Free State of Bavaria under the Excellence Strategy of the German Federal Government and the Länder. A.H. acknowledges funding by the Bavarian Hightech Agenda within the EQAP project.

## Author contributions

K.M., A.P., and Z.S. synthesized CrSBr crystals, and K.W. and T.T. provided hBN crystals. H.P.G.N. fabricated the field-effect devices. F.T.-V., H.P.G.N., and A.R. performed the experiments. A.S.B. developed the theoretical model with support from M.M.G. F.T.-V., H.P.G.N., A.R., P.M., M.M.G., A.S.B., and A.H. analyzed and discussed the data. F.T.-V., M.M.G., A.S.B., and A.H. wrote the manuscript. All authors commented on the manuscript.

## Funding

## Competing interests

The authors declare no competing interests.
