## [Peer Review File · Nature Communications]

Reviewers' Comments:

Reviewer #1:

Remarks to the Author:

The authors studied gate voltage-controlled excitons in layer-dependent CrSBr samples. Additionally, the authors used several approaches to justify experimental evidence, such as magneto-PL, reflectance, and Hamiltonian analyses. Furthermore, they established a mechanism for intra- and interlayer excitons, as well as intralayer and interband charge-bound states. The paper is abundant in experimental data, with well-organized figures and insightful interpretations of the experimental evidence. Hence, this paper is suitable for publication in Nature Communications without any doubt or need for changes, given its originality, organization, and novelty.

Reviewer #2:

Remarks to the Author:

In this manuscript, Farsane et al. studied the electric field effect of exciton and magnetic transitions in the bilayer and trilayer CrSBr. They observed exciton resonance energy and metamagnetic transition field changes with gate voltage and demonstrated that these unique properties can be used to visualize local magnetic order. The work is well-presented and the experimental results are convincing to me. Given the importance of electrostatic control of magnetism in 2D magnets for potentially energy-efficient spintronic applications, this manuscript deserves publication in Nature Communications.

The shortcoming of the present work is the electrostatic field and doping effects are always coupled in the current single-gated device structure. While I understand fabricating a dual-gated device and doing all the measurements again could require a lot of work, the authors should at least explain more clearly why the exciton resonance energy and metamagnetic transition field changes are attributed to be doping-induced.

Manuscript ID NCOMMS-23-62081-T

“Doping-control of excitons and magnetism in few-layer CrSBr”

by F. Tabataba-Vakili et al.

We thank the reviewers for their detailed assessment of our manuscript and their very positive comments. Reviewer #1 recommends the publication of our work as is. Reviewer #2 made an important remark with regard to the effects of electric field and electrostatic doping in single-gate devices, which we address in detail below. The respective changes to the manuscript are highlighted in the revised version in blue.

Response to Reviewer #2:

In this manuscript, Farsane et al. studied the electric field effect of exciton and magnetic transitions in the bilayer and trilayer CrSBr. They observed exciton resonance energy and metamagnetic transition field changes with gate voltage and demonstrated that these unique properties can be used to visualize local magnetic order. The work is well-presented and the experimental results are convincing to me. Given the importance of electrostatic control of magnetism in 2D magnets for potentially energy-efficient spintronic applications, this manuscript deserves publication in Nature Communications.

The shortcoming of the present work is the electrostatic field and doping effects are always coupled in the current single-gated device structure. While I understand fabricating a dual-gated device and doing all the measurements again could require a lot of work, the authors should at least explain more clearly why the exciton resonance energy and metamagnetic transition field changes are attributed to be doping-induced.

The Reviewer is absolutely correct in pointing out the shortcoming of single-gated field-effect devices with intertwined effects of doping and electric field. Being well aware of this shortcoming, we performed careful and elaborate studies on different positions of both samples to ensure that the main signatures reported in our work are indeed dominated by doping-control of excitons and magnetism. Our position-dependent studies shown in Fig. R1 confirm on the one hand that the electric field is indeed different in different sample positions due to the given single-gate device layout. On the other hand, and crucial to our study, we consistently found on both samples that electrostatic doping is the main effect for the voltage-controlled signatures of excitons and magnetism.

In particular, we observed at different positions of both bilayer and trilayer regions in both samples that the actual gate voltage for the formation of trions or Fermi-polarons is position-dependent, yet the binding energy of 19 meV and the doping-induced dispersion of the attractive and repulsive Fermi-polaron branches as key characteristics of doping are position-independent (see Fig. R1 d

Figure R1: **a – c**, Gate voltage sweeps on a bilayer region, recorded in absorption on two different positions of sample 1 (s1, pos. 1 and 2) and on sample 2 (s2), shown together with dispersions obtained from fits of the peak maxima in the absorption spectra. **d**, Dispersions of the attractive and repulsive polaron branches in the data of **a – c**, overlaid on the data in **a** by imposing lateral shifts of 5.5 and 1.2 meV on the data in **b** and **c**, respectively, with additional linear voltage scaling by 0.9 and 2.1. **e – g**, Same but for a trilayer region on two different positions of sample 1 (s1, pos. 1 and 2) and one position on sample 2 (s2). **h**, Overlaid dispersions with shifts of 3 and 1 meV for the data in **f** and **g**, respectively, with linear voltage scaling by a factor of 1.8 for the data in **g**. The horizontal lines in **d** and **h** mark the position-dependent onset of the attractive Fermi-polaron, which was used to determine the linear scaling parameter for the relative effective voltages at different positions. Lateral energy shifts of a few meV are most probably required due to slightly different exciton energies in different local dielectric environments.

and h). In particular, a comparison between a position on top of and near the graphene contact for the trilayer also shows identical dispersions (Fig. R1 e, f, and h). These findings, in agreement with related work on two-dimensional semiconductors (Ref. [31] K. F. Mak et al., Nat. Mater. 12, 207, 2013, and Ref. [32] J. S. Ross et al., Nat. Commun. 4, 1474, 2013), confirm that the main signatures of Fermi-polaron formation and the according dispersions of the attractive and repulsive polaron branches are dominated by doping.

With the confidence that the main exciton features are dominated by doping, we anticipate that doping also primarily dominates the magnetic response. This is consistent with the related studies on two-dimensional magnet CrI₃ (Ref. [1] B. Huang et al., Nat. Nanotechnol. 13, 544, 2018, and Ref. [2] S. Jiang et al., Nat. Nanotechnol. 13, 549, 2018), which showed that the critical magnetic field of the metamagnetic transition can be controlled by electrostatic doping. In particular, comparative studies of both single- and dual-gated devices ruled out significant contributions of the electric field to voltage-controlled magnetism in a single-gated device (Ref. [3] S. Jiang et al., Nat. Mater. 17, 406, 2018). The mechanism behind the doping-dependence is that the interlayer exchange energy is inversely proportional to the on-site Coulomb interaction, which in turn depends on doping (Ref. [2] S. Jiang et al., Nat. Nanotechnol. 13, 549, 2018, and Ref. [42] Y. Wang et al., Phys. Rev. B 108, 054401, 2023).

In response to the reviewer's comment, we have rephrased our statement regarding this effect in the main text to read: *"This effect, also observed in bilayer CrI₃ [1,2], is consistent with an inverse scaling of the interlayer exchange energy with doping, which reduces the critical field upon negative doping. According to the Kugel-Khomskii model, the interlayer exchange energy is inversely proportional to the on-site Coulomb interaction [42], which in turn is proportional to the electron density."*

With this additional insight and changes made to the manuscript in response to the comment of Reviewer #2, we hope that the publication of our work in Nature Communications can proceed with the support of Reviewer #2.

Reviewers' Comments:

Reviewer #2:

Remarks to the Author:

The authors have carefully addressed my concern. I recommend the publication of the current manuscript on Nature Communications.